# Addressing Chest Radiograph Projection Bias in Deep Classification Models

**Sofia C. Pereira**[1,2]                                    SOFIA.C.PEREIRA@INESCTEC.PT
[1] *Faculty of Engineering, University of Porto*
[2] *Institute for Systems and Computer Engineering,*
*Technology and Science, INESC-TEC*

**Joana Rocha**[1,2]                                       JOANA.M.ROCHA@INESCTEC.PT
**Alex Gaudio**[1,2,3]                                      AGAUDIO@ANDREW.CMU.EDU
[3] *Carnegie Mellon University*

**Asim Smailagic**[3]                                      ASIM@ANDREW.CMU.EDU
**Aurélio Campilho**[1,2]                                  CAMPILHO@FE.UP.PT
**Ana Maria Mendonça**[1,2]                                AMENDON@FE.UP.PT

**Editors:** Accepted for publication at MIDL 2023

## Abstract

Deep learning-based models are widely used for disease classification in chest radiographs. This exam can be performed in one of two projections (posteroanterior or anteroposterior), depending on the direction that the X-ray beam travels through the body. Since projection visibly affects the way anatomical structures appear in the scans, it may introduce bias in classifiers, especially when spurious correlations between a given disease and a projection occur. This paper examines the influence of chest radiograph projection on the performance of deep learning-based classification models and proposes an approach to mitigate projection-induced bias. Results show that a DenseNet-121 model is better at classifying images from the most representative projection in the data set, suggesting that projection is taken into account by the classifier. Moreover, this model can classify chest X-ray projection better than any of the fourteen radiological findings considered, without being explicitly trained for that task, putting it at high risk for projection bias. We propose a label-conditional gradient reversal framework to make the model insensitive to projection, by forcing the extracted features to be simultaneously good for disease classification and bad for projection classification, resulting in a framework with reduced projection-induced bias.

**Keywords:** chest X-ray, deep learning, disparity, anteroposterior, posteroanterior

## 1. Introduction

Artificial Intelligence (AI) revolutionized many fields, including medicine. It is crucial to ensure that AI-based methods are not only accurate but also unbiased and ethically compliant. Biased diagnoses can have serious consequences, and the potential sources of bias are countless, ranging from biological attributes (age, gender) to socioeconomic factors (race, income) and clinical factors (exam conditions, medical institution, presence of medical devices, among others). When AI models are trained on biased data, they become prone to shortcut learning: instead of learning meaningful features for the issue at hand, models base

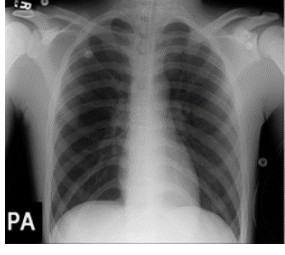 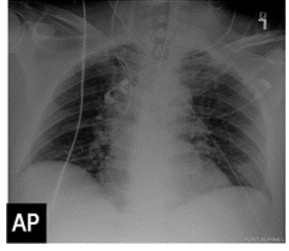

(a) Posteroanterior scan        (b) Anteroposterior scan

Figure 1: Examples of chest radiographs taken in both projections.

their predictions on biases in the data, which are typically easier to learn than meaningful features. Not only this poses serious ethical concerns but, consequentially, these models will fail in data where the same types of biases are not present.

In this paper, we focus on the bias that projection can inject in deep learning models for frontal chest X-ray (CXR) classification. These scans can be performed in one of two projections. In the posteroanterior (PA) projection (Figure 1a), the X-ray beam transverses the patient from posterior to anterior, which is the gold standard projection for frontal CXRs. However, it is not always possible to perform PA CXRs, especially when the patient is not able to stand. When that is the case, an anteroposterior (AP) projection (Figure 1b) is used, which does not require the patient to stand up, enabling the use of portable X-ray units. There are some key differences between both views. In the AP view, the scapulae are over the lung fields, which can interfere with the reading and interpretation of the scan. In the PA view, the scapulae are away from the lung fields. The clavicles are over the lung apex in the PA view, while in the AP they are above it. The heart appears enlarged in the AP view due to the shorter distance between the patient and the X-ray source. Finally, the appearance of the ribs also differs among projections, as the anterior ribs are clearer in AP and the posterior ribs are clearer in PA. Considering that these differences are immediately clear to a radiologist and possibly even to individuals with no training in radiology, a deep learning model can also easily learn them if that benefits its performance.

## 1.1. Related Work

Chest radiography data sets typically contain both PA and AP images (Irvin et al., 2019; Wang et al., 2017; Johnson et al., 2019), but may include exclusively PA images (Nguyen et al., 2022). Some authors choose to only include PA CXRs in their studies (Punn and Agarwal, 2021). This choice, motivated by the higher quality of the PA projection, severely limits the applicability of AI models in real clinical settings because only patients able to stand up will benefit from the algorithm to its fullest extent. The CXR projection may be a source of bias when correlations among projection and radiological findings exist in the data (Viviano et al., 2020), and especially when the bias patterns are easier to learn than the disease patterns. Kim et al. (2019) showed that CXR projection can be classified with very high accuracy, which further motivated our study.

Mehrabi et al. (2021) make an extensive review on bias and fairness in machine learning. They categorize bias into three types, according to its source: the data, the algorithm, and

the user. The bias that CXR projection may inject in disease classification models arises from the user and/ or the data. The user may create representation bias when collecting the data, generating spurious correlations between radiological findings and projection. The data itself may contain population bias when certain radiological findings are more common in a specific projection. For example, AP scans are typically more biased towards disease because they are only performed when a patient cannot stand.

Multiple approaches to mitigate or reduce bias in deep classification models have been proposed in the medical imaging domain, such as balancing the data according to the protected attribute(s) (Puyol-Antón et al., 2022), making protected attribute predictions close to the uniform distribution (Bevan and Atapour-Abarghouei, 2022) or training separate classifiers for each sub-group of the protected attribute(s) (Puyol-Antón et al., 2022). Feature disentanglement can be seen as one of the possible avenues to deal with biased models (Liu et al., 2022), where the goal is to generate a feature space where specific features can be associated with well-defined generative factors. For our application, this could be separating the feature space into disease-encoding and projection-encoding features. Gradient reversal, introduced by Ganin and Lempitsky (2014), can be seen as a way to achieve feature disentanglement. Initially introduced for adversarial domain adaptation but later adapted for bias removal (Raff and Sylvester, 2019; Ueda et al., 2021; Rundo et al., 2021), including in medical imaging applications (Bevan and Atapour-Abarghouei, 2022), this method enforces the feature spaces of both projections to overlap, resulting projection-agnostic features. It aims at generating predictive features that cannot be used to distinguish between different data domains, thereby allowing for effective data domain adaptation. The framework combines the optimization of underlying features with two discriminative classifiers: a label predictor that predicts the class labels and a domain classifier that predicts the source of the data. The goal of the network is to minimize the loss of the label predictor, while maximizing the loss of the domain classifier, such that the learned features are discriminative for the main task but not for domain classification. In the case of bias removal, instead of having a source and target domain, two sub-groups of a single domain are separable by a protected attribute to which models should be insensitive to, and the domain classifier is framed instead as a protected attribute classifier, whose loss shall be maximized.

## 1.2. Contributions

Our contributions are three-fold: (1) We show that projection is easier to learn than any of the radiological findings considered, even when a model is not trained for such a task. This puts models at high risk for projection bias, especially when projection and disease information correlate, so models tend to learn projection instead of meaningful disease features. (2) We also show that models for CXR classification are better at classifying images in the projection they were presented the most during training, suggesting that patient projection is a source of bias in the model's decisions and thus influences predictions, supporting (1). (3) We propose a debiasing approach that discourages the model from learning projection-relevant information when learning disease-relevant information, based on gradient reversal. Moreover, we show that by using the proposed approach, our model successfully *learns to ignore* projection, resulting in a less biased model that performs more equally between AP and PA projections.

## 2. Methods

### 2.1. Data and Preprocessing

We use the ChestX-Ray14 multi-label data set (Wang et al., 2017), which contains 112,120 frontal-view X-ray images of 30,805 patients, labeled for the presence of fourteen radiological findings and disease labels. The images are first resized such that the smaller edge is 256 pixels, preserving aspect ratio, and then cropped to $224 \times 224$ pixels. The data is then split into five stratified folds, without patient overlap, for five-fold cross validation, where four folds are used for training (89,696 $\pm$ 1 scans) and one for validation (22,424 $\pm$ 1 scans). Table 1 details the number of occurrences of each label in each fold and the percentage of PA images ($\%AP = 100 - \%PA$). Random horizontal flips ($p = 1\%$) and random rotations ($\pm 20°$ range) were performed to augment the data set.

Table 1: Number of occurrences per fold of each label and percentage of Posteroanterior (PA) occurrences.

| | Hernia | Cardiomegaly | Edema | Pneumothorax | Nodule | Emphysema | Infiltration | Atelectasis | Fibrosis | Consolidation | Mass | Pleural Thickening | Pneumonia | Effusion | No Finding |
|---|---|---|---|---|---|---|---|---|---|---|---|---|---|---|---|
| Occurrences | 45 ±10 | 555 ±7 | 461 ±3 | 1,060 ±6 | 1,266 ±9 | 503 ±6 | 3,979 ±7 | 2,312 ±9 | 337 ±3 | 933 ±8 | 1,156 ±11 | 677 ±5 | 286 ±6 | 2,663 ±6 | 12,072 ±1 |
| % PA | 85.28 ±4.11 | 56.33 ±3.96 | 11.98 ±1.47 | 64.26 ±2.36 | 65.98 ±2.55 | 59.59 ±3.74 | 47.01 ±1.32 | 49.55 ±1.26 | 83.52 ±1.56 | 32.58 ±1.36 | 61.68 ±1.37 | 71.43 ±2.01 | 44.05 ±3.56 | 49.48 ±1.52 | 65.11 ±0.70 |

### 2.2. Experiments

Two models were developed, both based on an ImageNet (Deng et al., 2009) pretrained DenseNet-121 (Huang et al., 2017) feature extractor, widely used in the literature for this data set (Rajpurkar et al., 2017; Baltruschat et al., 2019) and other CXR data sets (Irvin et al., 2019; Alam et al., 2022). Both models contain three modules: a **feature extractor** $\theta_{fe}$, a **disease classifier** $\theta_d$ and a **projection classifier** $\theta_p$ (Figure 2). $\theta_d$ contains 14 output neurons (one per disease label) and is used to predict a probability for each label $y_d = [y_1, y_{,2}, ..., y_{14}]$. The cross entropy loss between $\hat{y_d}$ and corresponding ground-truth $y_d$, $L_d$, is used to train $\theta_{fe}$ and $\theta_d$. $\theta_p$ and has a single output neuron for predicting the probability of an image being of the AP projection $y_{ap}$ ($P(PA) = 1 - P(AP)$). The cross entropy loss between $\hat{y}_{ap}$ and corresponding ground-truth $y_{ap}$, $L_p$, is used to update the parameters of $\theta_p$. In the baseline model (Figure 2a), the goal of predicting CXR projection is to assess how good the model is at predicting projection with features that are trained exclusively for disease classification, and hence, this neuron's output should not be considered when calculating the gradients for updating the feature extractor. To achieve this, the gradient back-propagated from the projection output neuron is not propagated beyond $\theta p$, preventing feature updates based on projection information. The total loss of the baseline model is the sum of $L_d$ and $L_p$, each referring to the parameters they update (Equation 1).

$$L = L_d(\theta_{fe}, \theta_d) + L_p(\theta_p) \tag{1}$$

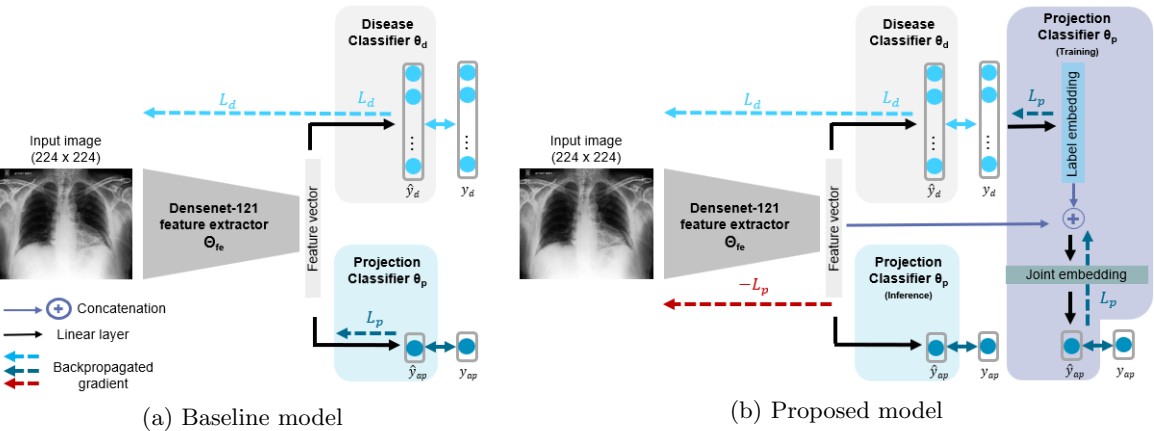

(a) Baseline model          (b) Proposed model

Figure 2: Forward and backward passes of both architectures, which share three main components: a feature extractor, a disease classifier and a projection classifier. The dashed arrows illustrate the parameters affected by each loss component from Eq. 1 and 2, respectively.

We then propose a modified version of the baseline model with the goal of reducing projection-induced bias (Figure 2b). To this end, we use gradient reversal (Ganin and Lempitsky, 2014) on the projection task to avoid the encoding of projection-related information in the features. There are two modifications to the baseline model. First, a new loss component is introduced, where the gradients calculated based on $L_p$ are inverted and used to update $\theta_{fe}$, to remove information encoding projection from the feature extractor. The proposed model plays a min-max game when updating $\theta_{fe}$, where it *learns to ignore* projection information while simultaneously learns to predict disease labels. The total loss of the proposed model therefore presented in Equation 2:

$$L = L_d(\theta_{fe}, \theta_d) + L_p(\theta_p) - L_p(\theta_{fe}) \tag{2}$$

The second modification takes place in $\theta_p$, where $y_d$ is encoded into an embedding that is concatenated with the feature vector. The concatenated vector is encoded into a joint embedding and used as an input to predict $\hat{y}_{ap}$ (Figure 2b). Therefore, the output neuron of $\theta_p$ is now generating $P(AP|y_d)$ instead of $P(AP)$. By using $P(AP|y_d)$ as $\hat{y}_{ap}$, instead of $P(AP)$, we account for the conditional dependence between projection features and disease labels when computing $L_p$, ensuring that the alignment between the AP and PA feature spaces is adapted according to the diseases present in each image. This conditional setting was inspired by Long et al. (2018), where the main task's outputs are concatenated with the feature representation and used as input for the domain classifier. We adapt this concept to CXR projection bias mitigation by conditioning projection classification to the ground-truth labels of the disease classification task. During inference, we do not use $y_d$ as an input for predicting $\hat{y}_{ap}$, and the model behaves exactly like the baseline model, in order to assess how well the model can classify projection independently of the disease, and to compare it to the baseline model.

We run both models (baseline and proposed) on the five-folded data. We report the Area Under the Receiver Operating Characteristic Curve (AUC) and for every disease label and

for the projection label. To measure bias, considering projection as the protected attribute of interest $P$, we calculate the True Positive Rate (TPR) Disparity between AP and PA images (Equation 3) (Raff and Sylvester, 2019; Mehrabi et al., 2021; Ruan et al., 2022; Rodolfa et al., 2021). This metric is based on the Equal Opportunity definition of fairness, introduced by (Hardt et al., 2016). This definition states that each group of $P$ should get a positive outcome at equal rates ($P(Y = 1|P = 0, Y = 1) = P(Y = 1|P = 1, Y = 1)$).

$$TPRDisp = |TPR_{AP} - TPR_{PA}| \tag{3}$$

Even though the goal is to obtain a model with less bias, predictive performance on the main task cannot be neglected. Motivated by this, and inspired by (Raff and Sylvester, 2019), we also report $\delta = AUC - TRPDisp$, in order to consider a trade-off between predictive performance and equality of opportunity.

## 2.3. Model Training Details

All the models were trained for eight epochs with a batch size of 32 and a learning rate of $1e^{-4}$ for the *learning* loss components $L_d(\theta_{fe}, \theta_d)$ and $L_p(\theta_p)$, and $1e^{-5}$ for the *learning to ignore* loss component $-L_p(\theta_{fe})$. The learning rates are reduced twice by a factor of ten after four and six epochs. Checkpoints are performed every 6,400 images. For the baseline, we select the weights from the checkpoint with the lowest $L_d$ validation loss, since $L_p$ does not affect the training of $\theta_{fe}$ nor $\theta_d$. For our proposed model, both $L_d$ and $L_p$ affect $\theta_{fe}$. However, their magnitude and decrease speed vary significantly, and finding a way to combine them into a single monitoring quantity is hard and empirical, a common problem in adversarial settings like ours (Zhan et al., 2018; Mao et al., 2022). For this reason, we select the weights from the checkpoint with the highest $\delta$ on the validation set. The models were created using the PyTorch framework and run on an NVIDIA GeForce RTX 3080 10 GB Graphics Processing Unit. Binary predictions are generated by thresholding based on the best operating point of the per-label Receiver Operating Characteristic curve.

## 3. Results and Discussion

We start by analyzing the results obtained with the baseline model, followed by the results obtained with the proposed model. Table 2 shows the results obtained with the baseline model for each label, as well as the macro-averaged results across disease labels and also the AUC for projection classification. For each label, predictive performance is worse in images from the projection that is less prevalent in the data (Table 1). For example, most of the "Hernia" cases are PA, and therefore its TPR is much higher in PA scans. The opposite happens for "Edema", which contains many more AP instances. For labels in which the proportion of AP and PA scans is balanced, the TPR Disparity is smaller ("Cardiomegaly", for example). Note that the AUC for projection classification is very high, although the feature extractor was not explicitly trained for this task. This indicates that predicting projection from disease-related features is extremely easy, putting the model at high risk for shortcut learning. Considering that the key differences between AP and PA scans (Figure 1) are easier to grasp to the inexperienced human eye than those of most disease labels, it is not surprising that deep learning models can easily distinguish between the two

Table 2: Mean and standard deviation 5-fold cross validation results of the baseline model (B) and the proposed model (P). The up/down arrow indicates that the higher/lower the metric, the better, respectively. Best results shown in bold. AUC: Area Under the Receiver Operating Characteristic Curve; TPR AP: True Positive Rate in Anteroposterior scans; TPR PA: True Positive Rate in Posteroanterior scans; TPR Disp: True Positive Rate disparity.

| | AUC ↑ (%) | | TPR AP ↑ (%) | | TPR PA ↑ (%) | | TPR Disp ↓ | | $\delta$ ↑ | |
|---|---|---|---|---|---|---|---|---|---|---|
| | B | P | B | P | B | P | B | P | B | P |
| Hernia | 90.32 ±2.46 | **90.63** ±2.94 | 55.0 ±16.24 | **80.56** ±7.08 | 81.89 ±6.84 | **83.1** ±7.58 | 27.56 ±21.34 | **3.21** ±4.23 | 62.77 ±19.96 | **87.42** ±4.61 |
| Cardiomegaly | **91.63** ±0.53 | 91.14 ±0.82 | **81.97** ±4.31 | 81.45 ±5.58 | **89.95** ±3.0 | 88.28 ±3.08 | 7.97 ±3.75 | **6.83** ±4.17 | 83.65 ±4.22 | **84.31** ±4.9 |
| Edema | **90.49** ±0.37 | 87.58 ±1.52 | **89.75** ±2.43 | 81.96 ±3.81 | 52.22 ±10.49 | **68.8** ±9.95 | 37.53 ±9.21 | **13.17** ±7.8 | 52.96 ±9.22 | **74.42** ±8.18 |
| Pneumothorax | **88.85** ±0.96 | 88.12 ±0.95 | 71.01 ±4.84 | **74.72** ±2.07 | 83.07 ±2.99 | **83.98** ±1.22 | 12.06 ±4.21 | **9.26** ±1.81 | 76.78 ±5.03 | **78.86** ±2.43 |
| Nodule | **78.31** ±0.31 | 77.45 ±0.39 | 65.97 ±5.16 | **66.46** ±6.58 | **68.58** ±1.86 | 64.39 ±2.39 | **3.39** ±3.13 | 4.61 ±2.24 | **74.92** ±3.36 | 72.84 ±2.37 |
| Emphysema | **92.63** ±0.97 | 92.23 ±0.59 | 85.43 ±4.46 | **86.88** ±5.0 | 82.69 ±1.15 | **83.17** ±3.8 | **3.01** ±4.66 | 3.71 ±4.86 | **89.63** ±3.75 | 88.52 ±4.55 |
| Infiltration | **72.47** ±0.32 | 71.8 ±0.81 | **81.9** ±2.73 | 78.0 ±0.74 | 42.72 ±4.79 | **46.75** ±4.74 | 39.18 ±2.84 | **31.25** ±4.33 | 33.29 ±3.08 | **40.55** ±4.77 |
| Atelectasis | **82.56** ±0.21 | 81.87 ±0.33 | **82.12** ±2.16 | 78.6 ±1.32 | 73.91 ±2.73 | **74.59** ±2.47 | 8.21 ±2.26 | **4.01** ±2.32 | 74.34 ±2.46 | **77.86** ±2.48 |
| Fibrosis | **82.32** ±0.83 | 79.92 ±1.57 | 45.23 ±15.2 | **70.39** ±7.71 | **77.19** ±2.14 | 72.55 ±5.64 | 31.96 ±14.94 | **6.33** ±5.02 | 50.35 ±15.32 | **73.59** ±4.6 |
| Consolidation | **81.24** ±0.84 | 79.67 ±0.78 | **87.7** ±1.62 | 84.13 ±1.41 | 53.33 ±5.0 | **66.64** ±3.92 | 34.34 ±4.8 | **17.49** ±3.4 | 46.88 ±5.08 | **62.18** ±3.11 |
| Mass | **86.14** ±0.09 | 85.81 ±0.44 | **71.93** ±3.74 | 69.63 ±6.06 | **80.45** ±3.11 | 76.22 ±3.6 | 8.52 ±4.03 | **6.92** ±6.55 | 77.61 ±3.94 | **78.89** ±6.23 |
| Pleural Thickening | **81.35** ±0.67 | 79.97 ±1.52 | 65.75 ±6.65 | **71.01** ±2.07 | **80.0** ±4.13 | 72.29 ±2.36 | 14.25 ±4.67 | **2.82** ±2.36 | 67.1 ±4.43 | **77.15** ±2.23 |
| Pneumonia | **76.89** ±1.11 | 76.57 ±0.95 | 78.78 ±3.2 | **79.24** ±5.91 | 54.51 ±4.25 | **55.05** ±6.47 | 24.27 ±3.6 | **24.2** ±4.46 | **52.62** ±4.27 | 52.37 ±4.17 |
| Effusion | **88.81** ±0.43 | 88.52 ±0.47 | **84.2** ±2.03 | 81.93 ±1.93 | **82.74** ±1.33 | 81.05 ±2.44 | 2.32 ±1.46 | **1.89** ±0.47 | 86.49 ±1.39 | **86.63** ±0.69 |
| Macro average | **84.57** ±0.2 | 83.66 ±0.66 | 74.77 ±1.5 | **77.5** ±1.24 | 71.66 ±1.26 | **72.63** ±1.18 | 18.19 ±2.05 | **9.69** ±1.08 | 66.39 ±1.97 | **73.97** ±1.4 |
| Projection AUC ↓ | 99.28 ±0.21 | **61.18** ±3.16 | | | | | | | | |

types of scans. These results highlight the need to take CXR projection into account when creating or choosing a data set for training a machine learning model. For example, a model trained on a data set containing mostly AP scans may perform well in an intensive care unit setting, where most patients will be in a critical state and won't be able to stand up, which means most scans will be taken using the AP projection. However, the same model will most likely perform poorly in a triage setting, where most patients are able to stand, and thus a PA projection will be preferred. Even though a model that learns statistical priors derived from these types of correlations may be correct most of the time, especially in the case of population biases (Section 1.1), such a model would not be making decisions based on disease features, but rather based on features of other confounding factors. If this correlation is removed, the model will most likely fail because it never actually learned what diseases look like.

The proposed framework aims at learning features that do not encode projection information, and thus it is expected that the projection AUC decreases when compared to the

baseline model. Results show that is the case, with projection AUC dropping from 99% to 61% (Table 2). To investigate if the features from the proposed model are more well aligned between the two groups of scans (AP and PA), t-distributed Stochastic Neighbor Embedding (t-SNE) plots (van der Maaten and Hinton, 2008) of the features of the baseline and proposed model for one fold were generated and can be found in Figure 3. There is a clear separability among the features of AP and PA instances for most data points in the baseline features (Figure 3a), indicating that projection is being encoded into the disease features. However, in features of the proposed model (Figure 3b), the AP and PA feature spaces are overlapped. These results suggest that the proposed technique can be used to regularize the features of models trained in data sets with high projection imbalance, but that are meant to be used in a scenario where both AP and PA settings are present.

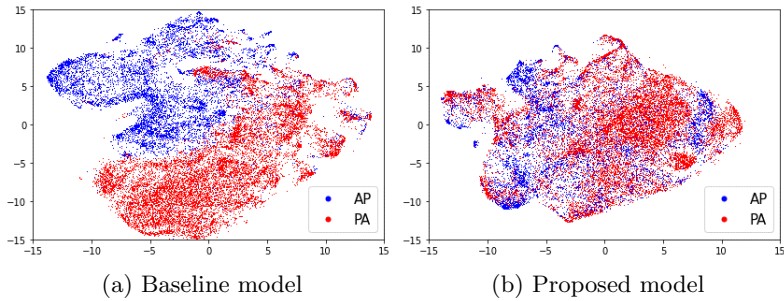

(a) Baseline model        (b) Proposed model

Figure 3: t-distributed Stochastic Neighbor Embedding plots showing the 2-D projected features.

The proposed model shows a decrease in the TRP disparity (Table 2), especially in labels with the highest projection imbalance, indicating that the equality of opportunity among AP and PA scans increased. Overall, there is a trade-off between predictive performance (AUC) and disparity. This is expected especially for labels with high projection imbalance because forcing the model to perform equally among both projections will make it perform better in scans from the underrepresented projection and worse in those from the overrepresented projection, leading to a decrease in the global label AUC and an increase in TPR parity.

## 4. Conclusions

Bias in medical AI can have devastating consequences. Projection is a potential bias source for CXR classification models. We showed that not only is projection very easy to learn, but also that models have higher TPR in scans from the most representative projection in the data set, and that projection is being encoded in the features. Finally, a strategy to address this problem was presented and discussed. We conclude that the projection of CXRs in a given data set needs to be taken into account by machine learning practitioners, as it may significantly affect model performance. Further work will focus on exploring the use of this methodology in other data sets and on jointly debiasing models for other attributes, whether they are clinical (pose, hospital department), acquisition-related (scanner model, image markers), biological attributes (age, gender) or socio-economical (race, insurance policy).

## Acknowledgments

This work was supported by the ERDF - European Regional Development Fund, through the Programa Operacional Regional do Norte (NORTE 2020) and by National Funds through the FCT - Portuguese Foundation for Science and Technology, I.P. within the scope of the CMU Portugal Program (NORTE- 01-0247-FEDER-045905) and LA/P/0063/2020. The work of S. C. Pereira was supported by the FCT grant contract 2020.10169.BD. The work of J. Rocha was supported by the FCT grant contract 2020.06595.BD. We thank José Pedro Gomes (University of Lisbon) for his help with the t-SNE plots.

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

## Appendix A. Limitations

ChestXRay-14's ground-truth labels are known to suffer from label inaccuracies due to the fact that these were automatically extracted using natural language processing. Moreover, the small resolution used ($224\times224$) may hinder the detection of some smaller abnormalities.

Although there is no reason to perform an AP scan when an orthostatic position is possible, due to their subpar quality when compared to PA scans, the data set does not detail if the AP scans are in fact supine (i.e., laying down) or semi-erect. In the case of semi-erect AP scans, some key differences among the projections described in Section 1 may not be present.

