# OpenReview forum: "Addressing Chest Radiograph Projection Bias in Deep Classification Models"
_MIDL.io/2023/Conference — MIDL 2023 Poster_

### Official Review · Reviewer_8fWV · 2023-01-29

**Confidence:** 4
**Preliminary Rating:** 5
**Recommendation:** Best Paper Award, Oral

**Summary:**

Deep learning models are notorious for picking up on spurious correlations that might not be useful features and are prone to short-cut learning. This work specifically focuses on addressing such short cut learning in chest X-ray imaging, where the projection bias (AP/PA) can be related to disease classification. This work systematically studies if the disease classification models are also learning projection labels, and then uses gradient reversal-inspired method to alleviate this bias. The experiments on ChestX-Ray 14 dataset is used along with DenseNet-121 type network, and demonstrate that the projection bias can be counteracted.


**Strengths:**

* The key issue addressed in this work -- shortcut learning and its impact on bias -- is an important one. The problem identified in CXR projection bias and the methods developed are relevant.


* Experimental evaluations are clear, and capture the influence of projection in classification in the baseline model quite clearly. That the AUC of projection classification in the baseline model is about 99\% is a remarkable result.

* Use of gradient reversal based methods to _unlearn_ projection information is a simple and reasonable strategy. And reducing the projection classification with this strategy addresses the problem identified in this work.

* I thoroughly enjoyed reading this paper; not just for its clarity in writing but also for the problem addressed along with the methods used.

**Weaknesses:**

I don't have any major concerns with this version of the work. This is a well written paper that addresses an important issue in a sound manner. Some discussions perhaps about how this could be used to counteract other biases can be useful. Also, any limitations of the current approach can be discussed.

**Deanonymize Review:**

no

**Detailed Comments:**

See review above.

**Paper Type:**

both

**Questions To Address In The Rebuttal:**

As I already mentioned above, there are no major concerns with this paper. Perhaps further discussion of extensions to other biases and limitations in the current study will make for a more balanced paper.

---

### Official Review · Reviewer_JZQ3 · 2023-02-02

**Confidence:** 5
**Preliminary Rating:** 3

**Summary:**

This paper investigates models trained to classify disease labels on chest X-ray and how they are biased based on the projection type of the X-ray image (postero-anterior(PA) or antero-posterior(AP)).  The paper demonstrates that there is a bias related to the projection type and that this can be corrected for during training by forcing the model not to learn features related to projection type.  Experiments are performed on 5 fold splits of the well-know public Chest-X-ray14 dataset.


**Strengths:**

The paper is well written in clear English and explanations and figures are generally sufficient.  There is clear motivation to reduce undesirable bias in AI models.  The experiments are clear, although some additional clarifications are suggested below.

**Weaknesses:**

There are several areas where clarification or additional information is required.  (See detailed comments below).  The authors use a very standard architecture and a fixed set of hyperparameters on images that are very downsampled and known to suffer from label inaccuracies, however the main point of the paper is not the classification performance.

**Deanonymize Review:**

no

**Detailed Comments:**

It is not clear to me that the methodology is novel, since the description of related work sounds as though it used the same method described here.  However, in the methods section it is also not explicitly stated that this method is drawn from the literature.  Please clarify whether this method has been used in previous work and use the appropriate citation in the methods section if so.  Otherwise it should be clarified how the method is novel compared with previous work.

It is clear how the five folds were derived from the CXR14 dataset, however it is not stated how these five folds are used in each experiment (e.g. 3 folds for training, 1 for validation, 1 for testing?).  Please state this explicitly, preferably with numbers/table to save the reader doing the mental math.  Numbers in test sets should also be provided in results tables.

It is not stated how hyper-parameters are chosen, and when the authors mention they choose the model checkpoint with the highest 'delta' value, it is not clear on which set they evaluate this - on a validation set?

Captions should explain acronyms.

It is not clear to me why TPR is used as the metric to compare performance across AP and PA subsets.  TPR as a standalone metric is not useful (since a model could simply output "positive" to all samples and obtain very high TPR, but this would be a useless model).  Why not use AUC on the subsets instead?

Please state how the image cropping is done, since without e.g. lung segmentation it is entirely possible to crop out part of the lungs.

By cropping the images to 224x224 it is impossible to detect smaller abnormalities, such as subtle nodules for example.  Also the labels of the CXR14 dataset are well known to suffer from inaccuracies.  These do not affect the overall message of the paper, but should be stated as limitations if time allows.

As a more general point, I wonder whether projection bias is always a bad thing.  Does a radiologist suffer from projection bias - or indeed all kinds of other bias?  If you know the patient cannot stand up for a PA image, then it is logical to be more likely to consider diagnoses which might go along with that.  Similarly if you know the patient is young/old, has a history of smoking, lives in a country where tuberculosis is endemic,... these are all factors that a radiologist considers when reviewing an image.   Further, if you think that the heart looks enlarged, then it might even be rather helpful to know if you look at a PA or AP image, since as the authors mention, the heart appears larger in AP view.    It is clear that models will perform worse on PA images if they have only ever seen AP images during training (and vice versa), and in such a case it may be wise not to allow the model to use features of projection.  However, if a model has been exposed to equally large numbers of AP and PA images for a particular label, could it be useful to allow it to also use projection information as a clue to diagnosis?   I would be curious to hear the authors thoughts on these things, and maybe a few lines in the Discussion to address it.

**Paper Type:**

validation/application paper

**Questions To Address In The Rebuttal:**

I would like the authors to address all points listed in Detailed Comments in the manuscript.   The only exception is the final point, about whether such bias is always a bad thing.  For that point I would like to hear the authors viewpoint in the rebuttal, and it may or may not contribute to a final change in the manuscript.

---

### Official Review · Reviewer_48gB · 2023-02-03

**Confidence:** 5
**Preliminary Rating:** 1

**Summary:**

The work studies the impact of the CXR projection on the prediction performance of deep learning based classifiers. The experiments isolate PA from AP and show results from each. The work also employs the gradient reversal (adversarial training) method to remove the difference in representations between the groups to try to avoid spurious correlations.

**Strengths:**

The motivation is interesting and the use of gradient reversal makes a lot of sense for the experiment in order to allow the model to learn from all the data while ignoring the view. .................

**Weaknesses:**

The experiment does not account for if the patient is laying down or not when selecting the AP view. AP itself is the direction of the beam but they can be distinguished into AP supine and AP erect. So while the experiments are interesting it is not comprehensive and this limits the contribution.

**Deanonymize Review:**

no

**Detailed Comments:**

> we pioneer the study of the effect of CXR projection in classification models.

This statement is incorrect. This other work looks at spurious correlations of AP vs PA in models and explores the same gradient reversal method among others: "Saliency is a Possible Red Herring When Diagnosing Poor Generalization" https://arxiv.org/abs/1910.00199


With the availability of so many large public datasets, why did this research not evaluate the models on an external dataset to better assess the generalization of the models?

I would take a look at the PadChest dataset as it contains annotations for AP Supine and AP views https://arxiv.org/abs/1901.07441

Here are some counts from the dataset:
```
PA           41675
L            18485
AP Supine     4633
AP            1301
COSTAL         149
```

> models perform better in scans from the most representative projection in the data set, indicating that projection is being encoded in the features and spuriously used to classify diseases

I do not believe there is an experiment to support this statement in the conclusion. For this there should be models trained on different proportions of views. In the models trained for Table 2 and 3 the same proportion of samples are used for both.


**Paper Type:**

validation/application paper

**Questions To Address In The Rebuttal:**

Add a limitations section outlining the limits of the conclusions based on the assumption of the position of the patient. Or conduct more experiments to remove this limitation.

Correct statements such as "a topic that had yet to be explored" as there exists existing literature.

Remove statements that are not supported by experiments such as about the representation of views in a dataset.

---

### Official Review · Reviewer_uXdz · 2023-02-04

**Confidence:** 4
**Preliminary Rating:** 4
**Recommendation:** Poster

**Summary:**

The authors investigate the effect of chest radiograph projection on the performance of deep learning-based classification models and propose a method to mitigate projection-induced bias in a supervised manner.
The initial results show that a DenseNet model is better at classifying images from the most representative projection in the dataset, suggesting that projection is taken into account by the classifier.
They propose a label-conditional gradient reversal framework to make the model agnostic to projections by forcing the extracted features to be simultaneously good for disease classification and bad for projection classification.

**Strengths:**

Generally, this work has a clear motivation, and it is written very well. The main strengths are:
1. This is a very good study to address bias in the classification model in the context of chest radiographs.
2. Good analysis of the results before and after the proposed method to remove bias.



**Weaknesses:**

I have some concerns:

**The term 'unlearning' seems to be incorrect, considering the training strategy.** From my understanding, 'unlearning' in the ML community is referred to as 'treatment,' i.e., when there is a **trained** model with potential bias or other issues, one proposes methods to correct/treat such a model. Differently, what the authors propose is a one-stage method that directly avoids bias from the beginning. This is **not** 'unlearning.'

**The methodology contribution is limited.** The reasons are as follows:
[1] The authors use labels of the projection to supervise the model and enforce the feature disentanglement. They compare it with a simple baseline (DenseNet with a feature discriminator). The gradient loss indeed helps to reduce the bias and does the job. But what I miss here is some discussion on feature disentangling methods that could enforce the model to disentangle the feature space into anatomy and disease subspaces.
[2] Following [1], since the labels are available, I wonder if performing sampling based on the project distribution would be helpful. Maybe the author could comment on this.

**Related work can be improved.** Since this work focuses on bias reduction, it would be appreciated if the author could discuss more related work on this topic in the medical image analysis community, such as [1]


**Table 2 and Table 3 can be merged, and the results from the two tables could be highlighted for the readers.**


References:

[1] Fairness in cardiac magnetic resonance imaging: assessing sex and racial bias in deep learning-based segmentation





**Deanonymize Review:**

yes

**Detailed Comments:**

Same as above


I have some concerns:

**The term 'unlearning' seems to be incorrect, considering the training strategy.** From my understanding, 'unlearning' in the ML community is referred to as 'treatment,' i.e., when there is a **trained** model with potential bias or other issues, one proposes methods to correct/treat such a model. Differently, what the authors propose is a one-stage method that directly avoids bias from the beginning. This is **not** 'unlearning.'

**The methodology contribution is limited.** The reasons are as follows:
[1] The authors use labels of the projection to supervise the model and enforce the feature disentanglement. They compare it with a simple baseline (DenseNet with a feature discriminator). The gradient loss indeed helps to reduce the bias and does the job. But what I miss here is some discussion on feature disentangling methods that could enforce the model to disentangle the feature space into anatomy and disease subspaces.
[2] Following [1], since the labels are available, I wonder if performing sampling based on the project distribution would be helpful. Maybe the author could comment on this.

**Related work can be improved.** Since this work focuses on bias reduction, it would be appreciated if the author could discuss more related work on this topic in the medical image analysis community, such as [1]. Disentangled learning should also be discussed (if the authors feel appropriate ).


**Table 2 and Table 3 can be merged, and the results from the two tables could be highlighted for the readers.**



References:

[1] Fairness in cardiac magnetic resonance imaging: assessing sex and racial bias in deep learning-based segmentation


**Paper Type:**

validation/application paper

**Questions To Address In The Rebuttal:**

The use of the technical term 'unlearning' should be commented on. It seems to be incorrect, considering the training strategy.

Clarification of methodology contribution.

Discussion on related work in the medical image analysis community.

---

### Official Review · Reviewer_Jb3m · 2023-02-06

**Confidence:** 4
**Preliminary Rating:** 2
**Recommendation:** Poster

**Summary:**

The aim of this paper is to examines the influence of chest radiograph projection on the performance of deep learning-based classification models and proposes an approach to mitigate projection-induced bias including two projections (posteroanterior or anteroposterior).
This is quite interesting topic. However, there should be too many thinings not to train project but to train compoundings like C-line, medical devices, L/R marks, etc. There is a lack of external validation, originality, etc.

**Strengths:**

Not only is projection very easy to learn, but also that models perform better in scans from the most representative projection in the data set, indicating that projection is being encoded in the features and spuriously used to classify diseases. Finally, a strategy to address this problem was presented and discussed.

**Weaknesses:**

However, there should be too many thinings not to train project but to train compoundings like C-line, medical devices, L/R marks, etc. There is a lack of ablation study, external validation, originality, etc.

**Deanonymize Review:**

yes

**Paper Type:**

validation/application paper

**Questions To Address In The Rebuttal:**

However, there should be too many thinings not to train project but to train compoundings like C-line, medical devices, L/R marks, etc. There is a lack of ablation study, external validation, originality, etc.

---

### Meta-Review · Area_Chair_j92F · 2023-02-23

**Recommendation:** Accept (Poster)
**Confidence:** 4

**Metareview:**

This paper examines the influence of chest radiograph projection on the performance of deep learning-based classification models and proposes an approach to mitigate projection-induced bias.
Here I summarize the key contributions and limitations.

Pros:
Interesting motivation reduce bias and clinical significance is clearly shown;
Good analysis of the results before and after the proposed method to remove bias;

Cons:
Lack of methodology novelty;
The study of bias in this application is not comprehensive and many other factors need to considered;
Lack of external validation on large-scale public datasets;
Other concerns can be seen in reviewers’ comments

In summary, we have received very diverse reviewer comments. After reading the reviewers’ comments and authors’ feedback, the study of project-induced bias could be inspiring for other biases problems. Generally, I am fine with acceptance, but suggest the authors could improve the manuscript substantially based on the reviewers' comments.

---

### Meta-Review · Program_Chairs · 2023-03-01

**Recommendation:** Accept (Poster)
**Confidence:** 5

**Metareview:**

During the PC discussion, it was felt that this is an overall interesting application that could motivate further analyses, so the decision was to accept this as a Poster.